# Predatory journals: Perception, impact and use of Beall's list by the scientific community–A bibliometric big data study

Georg Richtig [1,2,3,4¤]*, Marina Berger[5], Max Koeller[6], Markus Richtig[5], Erika Richtig [5], Jörg Scheffel[3,4], Marcus Maurer[3,4]*, Frank Siebenhaar [3,4]*

**1** Otto Loewi Research Center, Pharmacology Section, Medical University of Graz, Graz, Austria, **2** Division of Oncology, Division of Oncology, Medical University of Graz, Graz, Austria, **3** Institute of Allergology, Charité–Universitätsmedizin Berlin, Freie Universität Berlin and Humboldt-Universität zu Berlin, Berlin, Germany, **4** Fraunhofer Institute for Translational Medicine and Pharmacology ITMP, Allergology and Immunology, Berlin, Germany, **5** Department of Dermatology, Medical University of Graz, Graz, Austria, **6** Diagnostic and Research Institute of Pathology, Medical University Graz, Graz, Austria

¤ Current address: Department of Internal Medicine, Division of Oncology, Medical University of Graz, Graz, Austria

* georg.richtig@medunigraz.at (GR); frank.siebenhaar@charite.de (FS); marcus.maurer@charite.de (MM)

**Data Availability Statement:** All relevant data are within the paper and its Supporting Information files.

## Abstract

Beall's list is widely used to identify potentially predatory journals. With this study, we aim to investigate the impact of Beall's list on the perception of listed journals as well as on the publication and citation behavior of the scientific community. We performed comprehensive bibliometric analyses of data extracted from the ISSN database, PubMed, PubMed Central (PMC), Crossref, Scopus and Web of Science. Citation analysis was performed by data extracted from the Crossref Cited-by database. At the time of analysis, Beall's list consisted of 1,289 standalone journals and 1,162 publishers, which corresponds to 21,735 individual journals. Of these, 3,206 (38.8%) were located in the United States, 2,484 in India (30.0%), and 585 in United Kingdom (7.1%). The majority of journals were listed in the ISSN database (n = 8,266), Crossref (n = 5,155), PubMed (n = 1,139), Scopus (n = 570), DOAJ (n = 224), PMC (n = 135) or Web of Science (n = 50). The number of articles published by journals on Beall's list as well as on the DOAJ continuously increased from 2011 to 2017. In 2018, the number of articles published by journals on Beall's list decreased. Journals on Beall's list were more often cited when listed in Web of Science (CI 95% 5.5 to 21.5; OR = 10.7) and PMC (CI 95% 6.3 to 14.1; OR = 9.4). It seems that the importance of Beall's list for the scientific community is overestimated. In contrast, journals are more likely to be selected for publication or citation when indexed by commonly used and renowned databases. Thus, the providers of these databases must be aware of their impact and verify that good publication practice standards are being applied by the journals listed.

## Introduction

In the last decades, authors submitted their work for peer-review to be published in a paper-based journal. Costs were covered by a scientific society behind the journal, subscription fees

**Funding:** Georg Richtig received funding from the Austrian Science Fund FWF (W1241) and the Medical University Graz through the PhD Program Molecular Fundamentals of Inflammation (DK-MOLIN). The funders had no role in study design, data collection and analysis, decision to publish, or preparation of the manuscript.

**Competing interests:** The authors have declared that no competing interests exist.

to the journal, and/or industry sponsoring [1]. Importantly, access to articles is restricted to subscribers and those who pay per article. In the open-access (OA) model, articles are freely available to everyone who wants to read them [2]. However, the most common model of open-access is that authors pay a publisher an article-processing fee for the publishing services of a journal–including expenses for editing, distributing, hosting the article and peer-reviewing an article–and therefore for the option to make their article globally available at no cost for the reader. The transition of the open-access model in science has been performed in many different ways with many different OA models including green OA, Diamond OA, Gold OA, Bronze OA and many more [3]. Lately, the number of journals that exploit the OA model, so called "predatory" journals, has markedly increased. Predatory journals are set out to publish as many articles as possible without providing any or little publishing services including peer review, primarily to maximize their revenues [4].

There are several problems associated with publishing in a predatory journal. These include, but are not limited to, the fact that the lack of peer-review bears the risk of publishing data that are not fully scientifically sound. Peer review, i.e. the evaluation of reports by one or more scientists with similar competencies, is an important instrument of self-regulation that helps to maintain scientific quality standards, prevent plagiarism, and improve the quality of scientific reporting. A negative peer review should prompt journals to ask authors to improve their report or to reject the paper. This does not happen with predatory journals. Secondly, predatory journals have low or no standards with respect to the reporting of conflicts of interests, which results in the risk that such conflicts are not or not fully made transparent. Thirdly, many predatory journals exist for only a short period of time, after which their content and published reports are no longer available, resulting in loss of information [5].

How can the scientific community identify predatory journals? Several lists of predatory journals have been developed and are available, of which Beall's list is probably the most widely known. The list of Jeffrey Beall–a university of Colorado librarian–lists "potential, possible, or probable predatory journals" [6,7]. Beall's list catalogues these journals and their publishers (https://beallslist.net/), with the aim of limiting their use by the scientific community.

It has been proposed that Beall's list is widely used, although it is controversy discussed and is not updated anymore [8–11]. However, the last version of Beall's list has never been fully characterized, nor has its impact been investigated in detail in a computational way. The majority of studies performed on Beall's list have limited their analyses either to only its listed journals, excluding listed publishers, investigated only a sample of the complete list or used one of the earlier versions of the list [8,12]. Although there has been some work done in analysing citations of articles from potential predatory journals, these analyses were limited to databases which had more rigorous inclusion criteria [13].

It remains, therefore, unknown how many journals on Beall's list are included in quality-driven databases such as PubMed, how many articles are published by the journals on Beall's list and how this number has changed over the past years, or how often articles published by journals on Beall's list are referenced by other publications. The impact of Beall's list and, by proxy, of predatory publishing on the scientific community is ill characterized.

The aim of this paper is to characterize Beall's list in total, to see where such journals/publisher came from and in which databases such journals/publisher were listed since such metrics are widely used by authors to decide where to publish. Secondly, we aimed to see how journals that being listed on Beall's list performed–in terms of article count–in comparison to journals being listed in the DOAJ over the recent years. And thirdly, we investigate the number of citations which articles of journals being listed on Beall's list received. These goals should be reached by a computational big data approach using multiple APIs to obtain information in a systematic and reproduceable way with the limitation of server sided data quality and data quantity.

Our current study reports the results of a computational big data approach that analysed the number of Beall's list journals that i) are listed in quality-driven databases, ii) use geographic locations in the journal title divergent from their actual location of residence, and iii) are actively publishing. Also, we investigated to what extent Beall's list influenced the number of published articles by its listed journals in comparison to journals listed in the directory of open access journals (DOAJ) between 2011 to 2018. Finally, we assessed how often the scientific community cited articles published by journals on Beall's list.

## Materials and methods

### Beall's list of potentially predatory journals and publishers

The most recent version of Beall's list of potentially predatory journals and publishers was retained from webarchive (https://web.archive.org; 30.07.2017). The list was initially stored at https://scholarlyoa.com and consists of two independent parts: (1) a list with standalone journals (n = 1,289) and (2) a list of publishers (n = 1,162).

Each list contains the name of the journal or the publisher and a hyperlink to its homepage. We accessed each journal's and publisher's website included in Beall's list over a twelve-month period (01.04.2018 to 01.04.2019) using the hyperlinks provided. For each hyperlink, we recorded whether it directed to a website or a HTTP response status code only. We recorded, whether the content of the website represented a journal or a publisher and whether an International Standard Serial Number (ISSN) in the format XXXX-XXXX for the journal(s) was provided. All recorded information was stored in an excel worksheet for further processing (Microsoft, Redmond, WA).

### ISSN confirmation check

To check the validity of the given ISSN (print and/or online), we performed a manual query at the International Centre for the registration of serial publications (CIEPS, also known as the ISSN International Centre; https://portal.issn.org), for the same twelve-month period (01.04.2018 to 01.04.2019). As an identifier for a journal's country of origin, we used the "country" entry of the ISSN database.

### Assessment of journals' article count via Digital Object Identifier

To assess the number of articles published by a journal (and publisher), we made use of the Crossref database. This information was accessed via the official Crossref REST API (https://api.crossref.org/) using the ISSN as a unique journal's individual identifier. Records included the number of Digital Object Identifiers (DOI), as a surrogate for a published article, the availability of an individual linked ISSN, the number of DOIs issued per year from 2011 to 2018, the journal's title and the publisher's name.

### Identification of journal on Beall's list indexed in Scopus, Directory of Open-Access Journals, Web of Science, PubMed, and PubMed Central

First, complete lists of journals indexed by the following databases were obtained (by downloading the list from the corresponding database): Scopus (downloaded from Elsevier's Scopus database; https://www.scopus.com; 18.06.2018), Web of Science (WoS; downloaded from Clarivate Analytics; Journal Citation Reports 2017; https://jcr.clarivate.com; 12.04.2018), the Directory of Open-Access Journals (DOAJ; https://doaj.org/csv; 11.04.2018); PubMed (acquired from the U. S. National Institutes of Health (NIH; ftp://ftp.ncbi.nih.gov/pubmed/J_Medline.txt; 15.12.2018),

and the U.S. National Library of Medicine's list of PubMed Central (PMC) participating journals (https://www.ncbi.nlm.nih.gov/pmc/journals; 15.12.2018).

To analyse which journals on Beall's list are indexed by these databases, we compared their validated ISSNs (via CIEPS; portal.issn.org) with journal titles on Beall's list using an R script. For this purpose, three algorithms were used: I) a highly strict mode by which only the following characters and words were removed from journal titles: "&" / "and" / "." / "," / "-"and ":" (highly strict comparison); II) a less strict mode that additionally removed ["the"] and everything between square and round brackets: "[. . .]" / "(. . .)" (Less strict comparison); and III) a string comparison method using the Levenshtein algorithm (R package: RecordLinkage). For the Levenshtein algorithm, we excluded terms that are commonly used in journal titles including publisher prefixes ("maws" / "hsoa" / "iosr" / "ijrdo" / "ipasj" / "ictact" / "isca" / "meritresearch" / "signpostopenaccess") and other frequently used title elements ("journal" / "journals" / "of" / "openaccess"). Levenshtein cutoffs were set manually to a threshold the first journal showed a significantly different title compared to Beall's list and for assessing a range in which the Levenshtein cutoff should be set within big data studies. Additionally, all entries were manually validated and Levenshtein cutoffs were set to 0.7 for ISSN, 0.5 for Crossref, 0.6 for DOAJ, 0.8 for PMC, 0.65 for Pubmed, 0.7 for Scopus, and 0.7 for WoS. As a last step, all results were cross-checked manually in a direct comparison of the database entries with the journal title as it appeared on Beall's list. Only manually confirmed matching journal titles were used for further analyses.

Databases were categorized as quality driven and non-quality driven databases depending on their journal inclusion criteria. If journals had only to fulfill formal criteria to be included in the database, the database was categorized in non-quality driven database (ISSN database, Crossref). If there has been a substantially background check performed by the database prior to inclusion the database has been categorized as quality-driven (WoS, PMC, Pubmed, Scopus, DOAJ).

### Citation analysis using crossref's Cited-by program

Citation analyses were performed by linking the Crossref database to citation data entries listed in the journals' Crossref's Cited-by metadata. All journals that deposit their references in the Crossref database (deposits.references.current) were included in the analysis.

### Computational analyses and graphing of results

Data were obtained manually or by using Bash (Bourne-again shell) scripts and tools, the R software environment for statistical computing and graphics (version 3.4.3.) and PHP (version 7.3.0beta1). Graphs were created by using ggplot2 (version 2.2.1.) and GIMP (version 2.10).

### Data and code availability statement

The data that supports the findings and the code that generated the data of this study are available from the corresponding author upon reasonable request.

### Statistical analysis

Baseline characteristics were reported as means and their standard deviation, as medians with interquartile range, or as percentages. The Kendall rank correlation was used to assess the relationship between article count and citation count, where normality has not been met (assessed by Shapiro-Wilk test).

Mann-Whitney U test has been performed for non-normally distributed continuous data. Kruskal-Wallis-Test with pairwise comparisons using Wilcoxon rank sum test with continuity correction, and post-hoc correction using Bonferroni's post-hoc correction was used to analyse continuous variables non-normally distributed with more than two groups to compare.

A two-sided alpha level of 0.05 was considered significant. Statistical analysis has been performed using R software (version 3.4.3.) and SPSS version 23.0 software (SPSS, Chicago, IL).

## Results

### Initial analysis of Beall's list and preprocessing

Beall's list of potentially predatory journals includes 1,289 journals and corresponding hyperlinks to their websites. Of these, almost all (n = 1,226, 95.1%) led to the journals' websites, 57 hyperlinks were not accessible or did not direct to a scientific journals' homepage, and 6 resulted in a virus warning. A total of 1,080 (83.8%) journal websites provided a valid ISSN (in the format: XXXX-XXXX) and were included in further analyses.

Beall's list also contains 1,161 publishers and corresponding hyperlinks. In 16 cases, these hyperlinks led to virus warning, and 390 hyperlinks led to websites that were not accessible (Website not reachable or website in Chinese or other non-Latin language or not a scientific publishers' website). The remaining 755 publisher sites listed 22,677 journals (Mean: 30; Median: 8; Min: 1; Max: 1.215; 172 publishers listed > 20 journals). Of those, 11,380 journals listed an ISSN number. Together with Beall's list of journals, the merged list–cleared for duplicates–contained 21,735 individual journal titles linked to 10,354 ISSN numbers (S1a Fig in S2 File).

### Journals on Beall's list are rarely listed in quality-driven databases

Of the 10,354 journals on Beall's list, the majority were listed in general databases such as ISSN (8,671; 83.7%) and Crossref (5,244; 50.7%), whereas a much smaller number was indexed in quality-driven databases such as PubMed (1,155; 11.2%), Scopus (583; 5.6%), DOAJ (226; 2.2%), PMC (136; 1.32%) or Web of Science (51; 0.5%), which perform a more thorough background check. In total, for 82.6% ISSN journals on Beall's list, no match was found with any of the investigated quality-driven databases.

However, when including the journal name into the similarity check, 8,266 of 8,671 ISSNs (95.3%) were linked to the ISSN database, 5,155 of 5,244 (98.3%) to Crossref, 1,139 of 1,155 (98.6%) to PubMed, 570 of 583 (97.8%) to Scopus, 224 of 226 (99.1%) to DOAJ, 135 of 136 (99.3%) to PMC and 50 of 51 (98.0%) to Web of Science (S1 Table in S3 File).

Of the Beall's list ISSN journals listed in any quality-driven database, 13.2% only appeared in a single one, 39.6% and 35.3% were listed in two or three, respectively (S1b Fig in S2 File). Only 23 of 8,266 ISSN journals on Beall's list (0.3%) appeared in all five databases.

All journals listed in PMC were also indexed by PubMed and the majority of ISSNs listed in PMC (126/135; 93.3%) and PubMed (899/1.004; 89.5%) proved to be valid ISSNs (Fig 1A). Since a quality marker for a good journal is the number and the quality of the databases it is listed in, an overlap analysis of the ISSN database, DOAJ, PubMed, Scopus and Web of Science revealed that none of the journals on Beall's list was listed exclusively in the Web of Science database. However, there were 15 ISSNs (6.7% of 224 all ISSNs) exclusively listed in the DOAJ, 58 ISSNs exclusively in Scopus (10.2% of 570) and 93 ISSNs exclusively listed in PubMed (8.2% of 1,139, Fig 1B).

### Common title components and country locations of journals on Beall's list

After cleaning the list for general journal title components (e.g. "journal", "and", "of") and reducing the list to the 50 most commonly used terms in journal titles, the most frequent

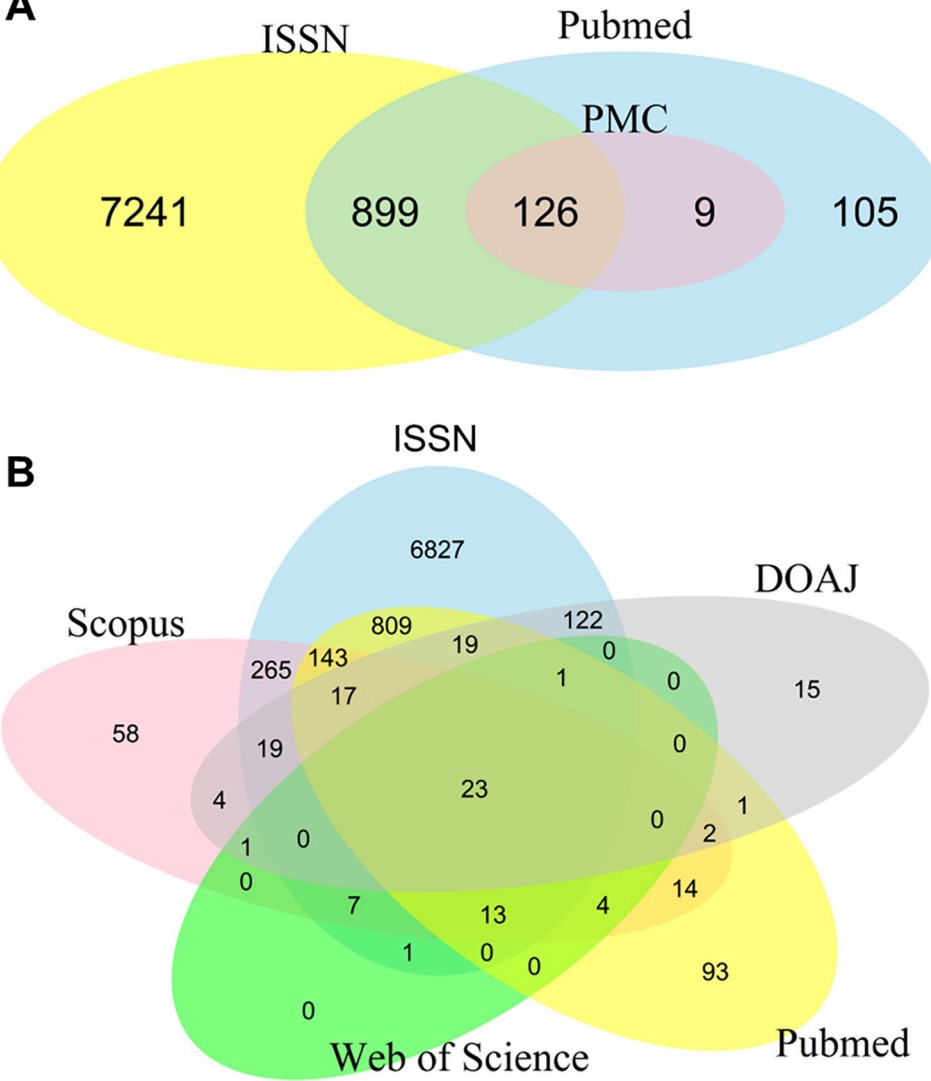

**Fig 1. Journals listed on Beall's list and their listing in different databases. A,** Venn diagram of ISSN overlaps between the ISSN database (yellow), PubMed (blue) and PMC (red) of all journals listed in Beall's list. **B,** Venn Diagram of all ISSNs listed in the ISSN database (blue), DOAJ (grey), PubMed (yellow), Web of Science (green) or Scopus (pink) of all journals listed on Beall's list.

journal title components were "open" (n = 5,216, 13.2%), "international" (n = 4,940, 12.5%), "science" (n = 2,355, 6.0%), "engineering" (n = 2,140, 5.4%) and "American" (n = 1,967, 5.0%; S1c Fig in S2 File).

One of the important questions, when it comes to a journal's credibility, is whether the geographic location indicated by the journal title matches its real location. When linking the ISSN country information with the journal titles, the majority of journals on Beall's list were registered in the United States (n = 3,206, 38.8%) followed by India (n = 2,484, 30.0%), the United Kingdom (n = 585, 7.1%) and Pakistan (n = 408, 4.9%) (Fig 2A and S1d Fig in S2 File). The journal title term "international" was mainly linked to India (n = 1,447, 51.5%) and the USA (n = 603, 21.5%), "American" was nearly exclusively linked to the USA (n = 285, 86.9%), "British" largely to the United Kingdom (n = 19, 76.0%) and "Asian" mostly to India and Pakistan (n = 79, 43.2% and n = 61, 33.3%, Fig 2B).

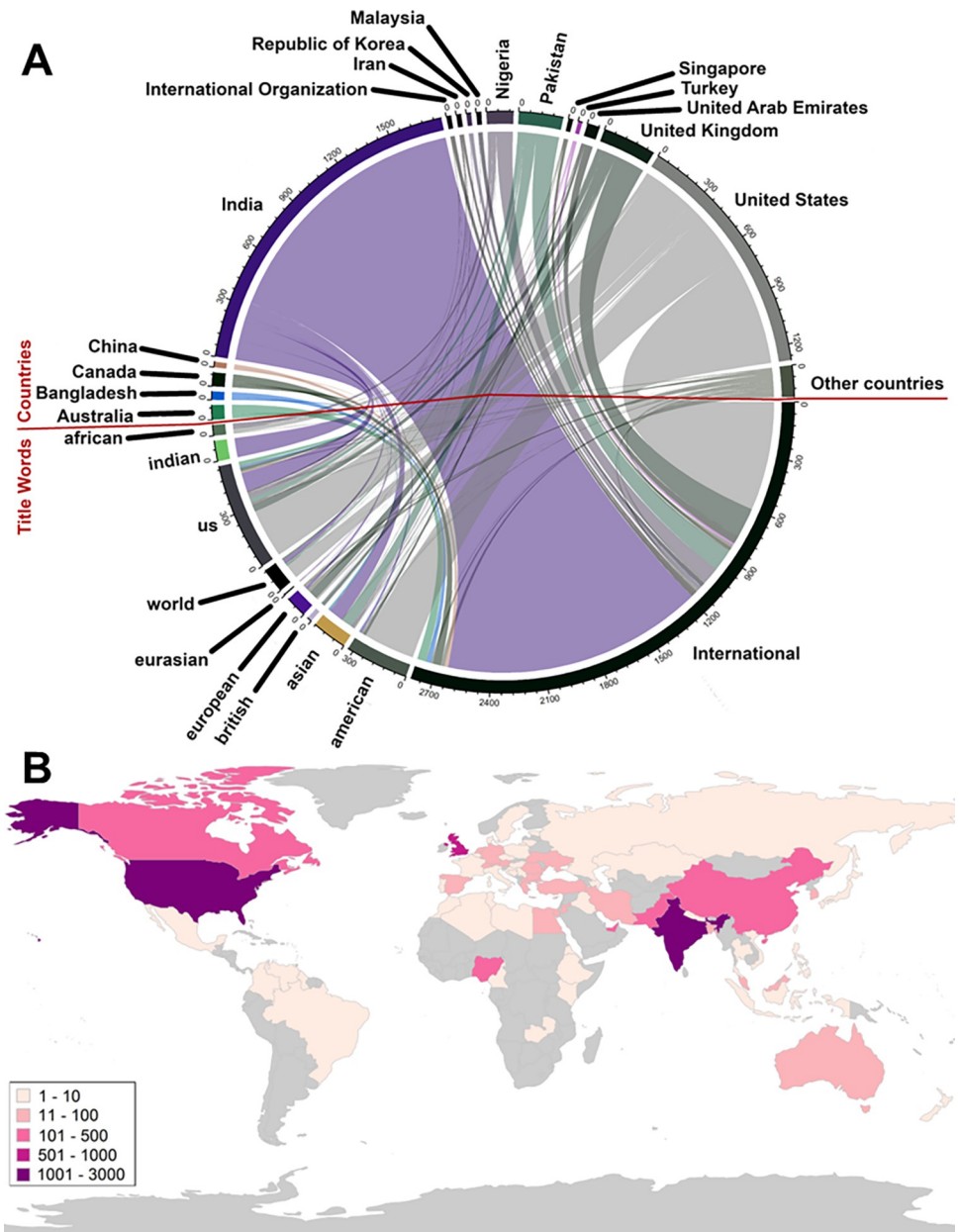

**Fig 2. Characterization of Beall's list in terms of country of origin and title geographical location. A,** Number of journals on Beall's list per country as provided by the ISSN database. **B,** Linking of commonly used geographical descriptions in journal titles of journals on Beall's list with country information provided by the ISSN database.

## The majority of journals on Beall's list are actively publishing

We measured the publication output of Beall's list journals between 2011 and 2018, by use of the DOIs registered on Crossref as a surrogate for published articles. Of 5,155 Beall's list journals in the Crossref database, 573 did not issue any DOIs. The remaining 4,582 journals continuously increased their number of articles published from 2011 to 2017, followed by a slight decrease in 2018 (Fig 3A).

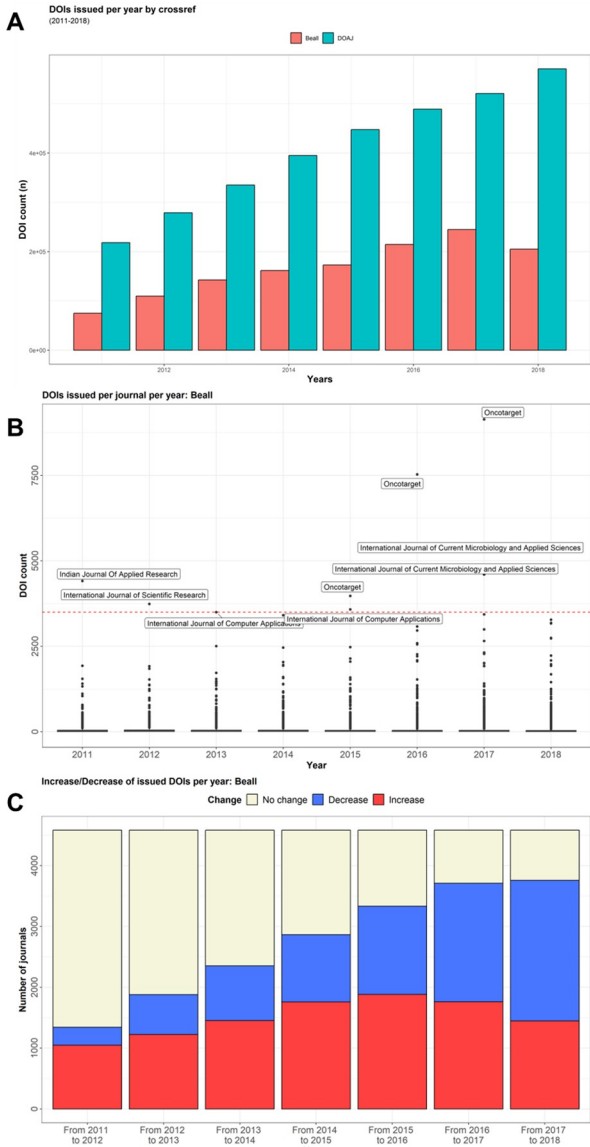

**Fig 3. Characterization of journals being listed on Beall's list and their development over time (2011–2018). A,** Issued DOIs from 2011 to 2018 of all journals listed on Beall's list in comparison to all journals listed in the DOAJ. Crossrefs data was used to retrieve the DOI count for each journal on Beall's list or the DOAJ and put in direct comparison for the period of 2011 to 2018. **B,** Box plot of all DOIs issued by crossref over the years of 2011 to 2018 to journals listed on Beall's list. **C,** Change of issued DOI count from year to year of journals listed on Beall's list. From crossref's data, it was calculated if a journal increased, decreased or had no change in their yearly article count compared to the previous year.

The *punctum maximum* of published articles per journal was reached in the year 2012 (Median: 24; Mean: 63.9; IQR: 45). The number of publishing journals increased to a maximum in 2017 (S2a Table in S3 File), whereas there was a significant increase in published articles from 2011 to 2012 (p = 0.00414) and a significant decrease from 2017 to 2018 (p<0.001)(Fully detailed in S2b Table in S3 File). The fastest growing journal was *Oncotarget*, which exceeded 3,500 issued DOIs per year in 2015 (Fig 3B). The total number of journals with growing numbers of articles increased until 2016 and declined thereafter. In 2018, more journals (50,4%) on Beall's list exhibited a decrease of published articles than an increase (31,6%) (Fig 3C).

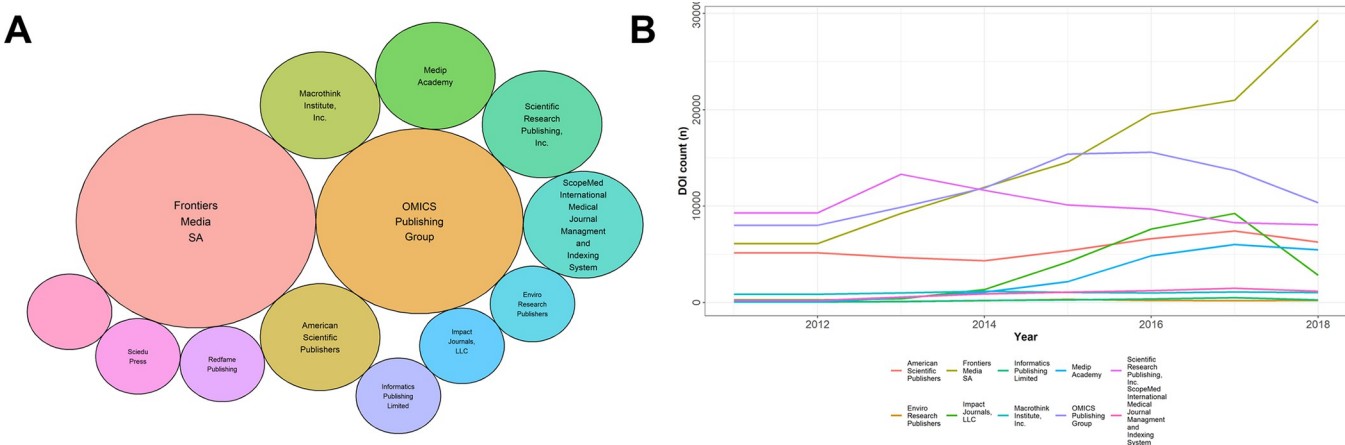

**Fig 4. Article count of publishers which were listed on Beall's list in the time period 2011 to 2018. A,** Bubble plot of all publishers listed on Beall's list, which had journals that had a continuous increase of issued DOIs over the last six to seven years. Size of bubbles represents the number of journals of the given publisher that had a continuous increase in issued DOIs over the last six to seven years as calculated from Crossrefs data. **B,** Line plot of the biggest publishers on Beall's list with the total number of issued DOI count during the years 2011 to 2018. Article count information (DOI count) was retrieved from Crossref's database and plotted for the period of 2011 to 2018 for the ten biggest publishers listed on Beall's list.

Nevertheless, between 2010 and 2018, four journals had a seven-year and 26 journals a six-year record of steadily increased article counts, respectively (n = 30/4,582; 0.7%; S3 Table in S3 File). During that time, eight journals showed a continuous decrease in their article count for seven consecutive years and 19 for six years (n = 27/4,582; 0.6%). Continuously growing journals were hosted by only 13 of 490 (2.7%) publishers on Beall's list, primarily Frontiers Media (eight journals) and OMICS Publishing Group (6 journals; Fig 4A, S3 Table in S3 File).

Publishers of continuously growing journals also published the largest number of articles between 2011 and 2018 (S2b Fig in S2 File). Frontiers Media issued 115,494 DOIs in 64 journals, followed by OMICS Publishing Group (87,511 DOIs, 460 journals) and Scientific Research Publishing (75,878 DOIs, 246 journals). Virtually all publishers experienced a decline in article counts after 2017, except Frontiers Media who continued to increase the number of published articles and showed a marked increase in 2018 (Fig 4B).

### Articles published in journals in Beall's list show similar publishing patterns to those in journals not included in the list, suggesting that Beall's list had no major influence on the Open-Access movement

To assess the impact of Beall's list on Open-Access publishing as well as on journals' publishing output of journals on Beall's list, we compared the article count of journals on Beall's list in comparison to journals on the DOAJ. Of 13,203 journals listed in the DOAJ, 7,876 were also listed in Crossref and, therefore, suitable for subsequent analysis. Similar to Beall's listed journals, open-access journals listed in the DOAJ exhibited a continuous increase in the number of issued DOIs between 2011 and 2018 (Fig 3A), whereas the total count of articles published by journals listed in DOAJ was always significantly higher than articles from journals on Beall's list (S2c Fig in S2 File). The maximum of published articles per journal was reached in the year 2013 (Median: 29; Mean: 69; IQR: 41). However, the mean number of articles published by journals has steadily increased over the entire period from 62 per year in 2011 to 81 per year in 2018 (S4a Table in S3 File). However, there was no significant increase in articles when compared to the previous year (Detailed in S4b Table in S3 File).

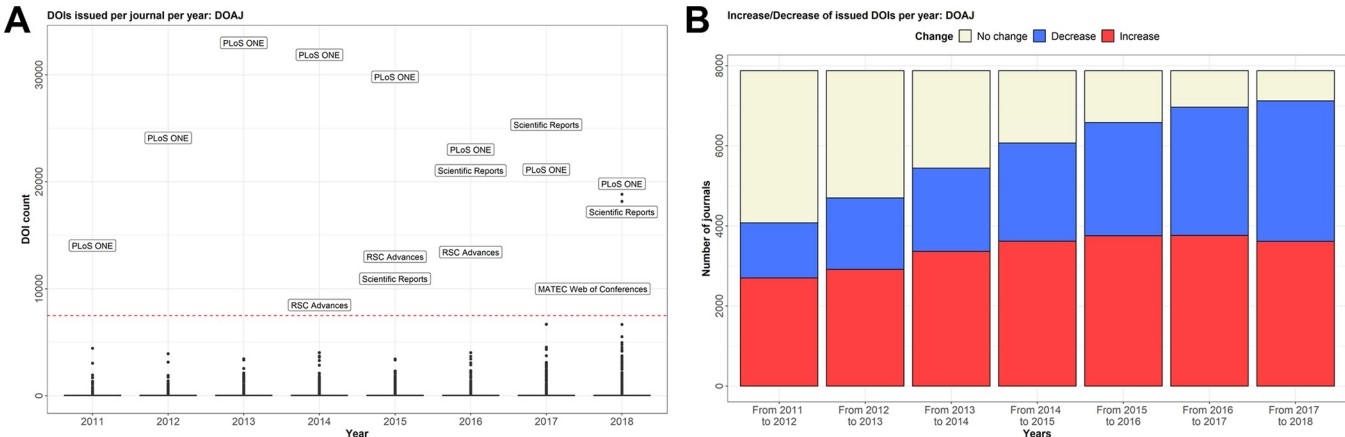

**Fig 5. Characterization of journals being listed in the DOAJ and their development over time (2011–2018). A,** Box plot of all DOIs issued by crossref over the years of 2011 to 2018 of journals listed in the DOAJ. All journals that had more than 7,500 DOIs issued per year were labeled with their respective journal title. **B,** Change of issued DOI count from year to year of journals listed in the DOAJ. From crossref's data, it was calculated if a journal increased, decreased or had no change in their yearly article count compared to the previous year.

Until 2006, *PLoS ONE* was the largest journal according to the number of published articles, with a maximum number of published articles in 2013 (n = 32,992) (Fig 5A). The number of journals with decreasing annual output increased continuously from 2011 to 2018 (Fig 5B).

When the journals' annual output was more closely analysed over the past seven years, 52 journals (0.7%) increased the number of published articles from year to year (median: 3), and no journal had a continuous decrease over the entire seven year period.

When journals were grouped by their publishers, 104 of 2,296 (4.5%) publishers (287 journals) had a continuous increase in published articles over the six (or seven) year period. Within the DOAJ listed journals, MDPI was the publisher with 74 journals with a continuous increase in published articles over this time, followed by Springer (Biomed Central Ltd.) with 21 journals (Fig 6A and S5 Table in S3 File). However, in terms of article count, Springer (Biomed Central Ltd.) ranked first, with 286,294 issued DOIs in 392 DOAJ listed journals, followed by

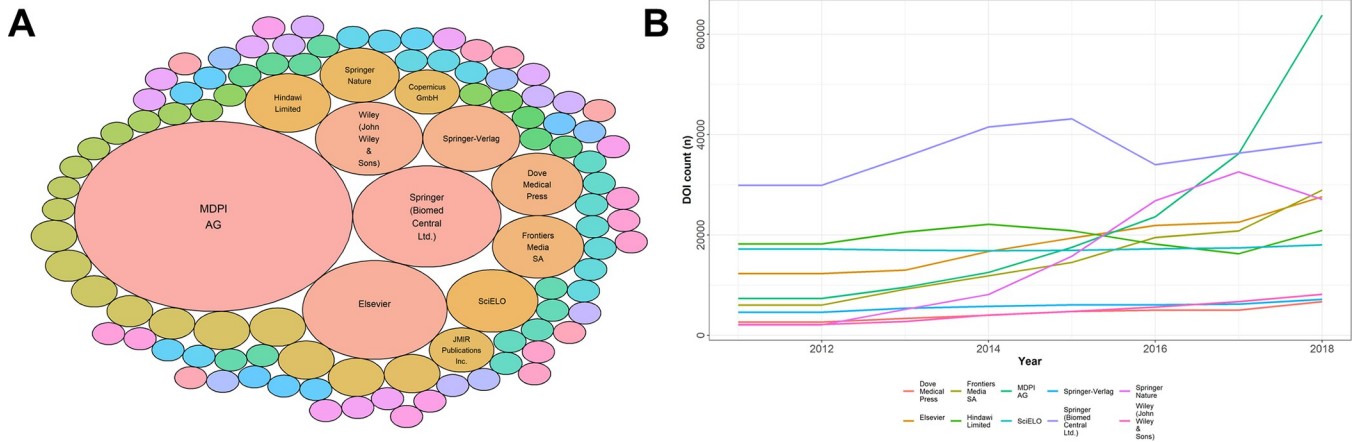

**Fig 6. Article count of publishers which were listed in the DOAJ in the time period 2011 to 2018. A,** Bubble plot of all publishers listed in the DOAJ who had journals that had a continuous increase of article publications over the last six to seven years. Size of bubbles represents the number of journals of the given publisher that had a continuous increase in issued DOIs over the last six to seven years as calculate from crossref's data. **B,** Line plot of the biggest publishers in the DOAJ with total number of issued DOIs during the years 2011 to 2018. Article count information (DOI count) was retrieved from Crossref's database and plotted for the period of 2011 to 2018 for the ten biggest publishers listed in the DOAJ.

Public Library of Science (PLoS) with 222,180 DOIs in 7 journals and MDPI with 175,717 DOIs in 155 journals (S2d Fig in S2 File).

When the total count of DOIs issued per publisher was divided by the number of years ranking within the top ten publishers—a measure for continuous development—Springer (Biomed Central Ltd.), Hindawi, SciELO and Springer-Verlag published at a consistent rate (Fig 6B).

### Journals on Beall's list get cited

An important question on the reputation of articles published by journals on Beall's list is whether they are recognized by the scientific community and cited by authors publishing in journals not listed by Beall. To address this question, we used Crossref's Cited-by service data [14].

5,145 journals on Beall's list were found in Crossref's database, 812 (15.8%) were actively contributing to the Cited-by service in the period of 2011 to 2018 (S2a Fig in S2 File). When combined with ISSN country data, we found that the majority of participants of the Cited-by program came from the United States (n = 359; 44.2%), followed by India (n = 101; 12.4%), Switzerland (n = 68; 8.4%), Pakistan (n = 62; 7.6%) and the United Kingdom (n = 44; 5.3%). On the other hand, the majority of journals not participating in the Cited-by program were from the United States (n = 1.817; 41.9%), India (n = 1.049; 24.2%), United Kingdom (n = 273; 6.3%), unknown (n = 273; 6.3%) and Pakistan (n = 62; 4.5%, S6 Table in S3 File).

Firstly, we linked Crossref's meta-data with Beall's list. Only 812 out of 5,244 (15%) Crossref-listed journals on Beall's list provided the required meta-data and were, thus, eligible for analysis. In total 3,796 journals (72,4%) were not listed in any other database (Scopus, WoS, PMC, PubMed, DOAJ) investigated here (S3A Fig in S2 File). The highest likelihood of Cited-By program participation was among journals listed in WoS (OR = 10.7; CI 95%: 5.5 to 21.5), followed by PMC (OR = 9.4; CI 95%: 6.3 to 14.1) and the DOAJ (OR = 5.8; CI 95% 4.2 to 8.1, S7 Table in S3 File).

Secondly, we investigated, which of the 812 Beall's list journals were cited by other journals. We excluded 59 journals that did not publish in the investigated time period between 2011 and 2018, leaving 753 journals for citation analysis. In total, these 753 journals published 358,736 articles (mean/median per journal: 476/150), which were cited 1,441,116 times (mean/median per journal: 1,913/133).

To investigate whether the inclusion in certain databases had an impact on citations, we categorized the 753 included journals as listed and non-listed in DOAJ, PMC, PubMed, WoS and Scopus.

Being listed in the DOAJ had the smallest impact on citations (Fig 7A), whereas being listed in WoS had the biggest positive impact (Fig 7B), and being listed in one of these databases had a higher correlation between citations and articles published than not being listed (Fig 3A and 3B; S3b-S3e Fig in S2 File). All other investigated databases including Pubmed (S3b Fig in S2 File), PMC (S3c Fig in S2 File) and Scopus (S3d Fig in S2 File) showed similar impact (databases commonly used to search for scientific articles). Therefore, we were interested to know whether a Beall's list journal, which is listed in these databases and has a higher likelihood of being found by researchers and thus has a higher likelihood of being cited. Journals listed in Pubmed, PMC or Scopus (n = 300) were cited 4,455 times, on average, as compared to 231 for journals not listed in any of these databases (n = 453) (S2d and S3e Figs in S2 File). Articles in Beall's list journals listed in PMC or WoS had significant higher median citation rates as compared to not-listed journals (Fig 8A), although this was shown to some extent for all other databases as well. Vice versa, articles with more than 50 (Fig 8B) or 100 citations (S3f Fig in S2 File) were significantly more frequently listed in PMC and WoS. Also, the percentage of articles

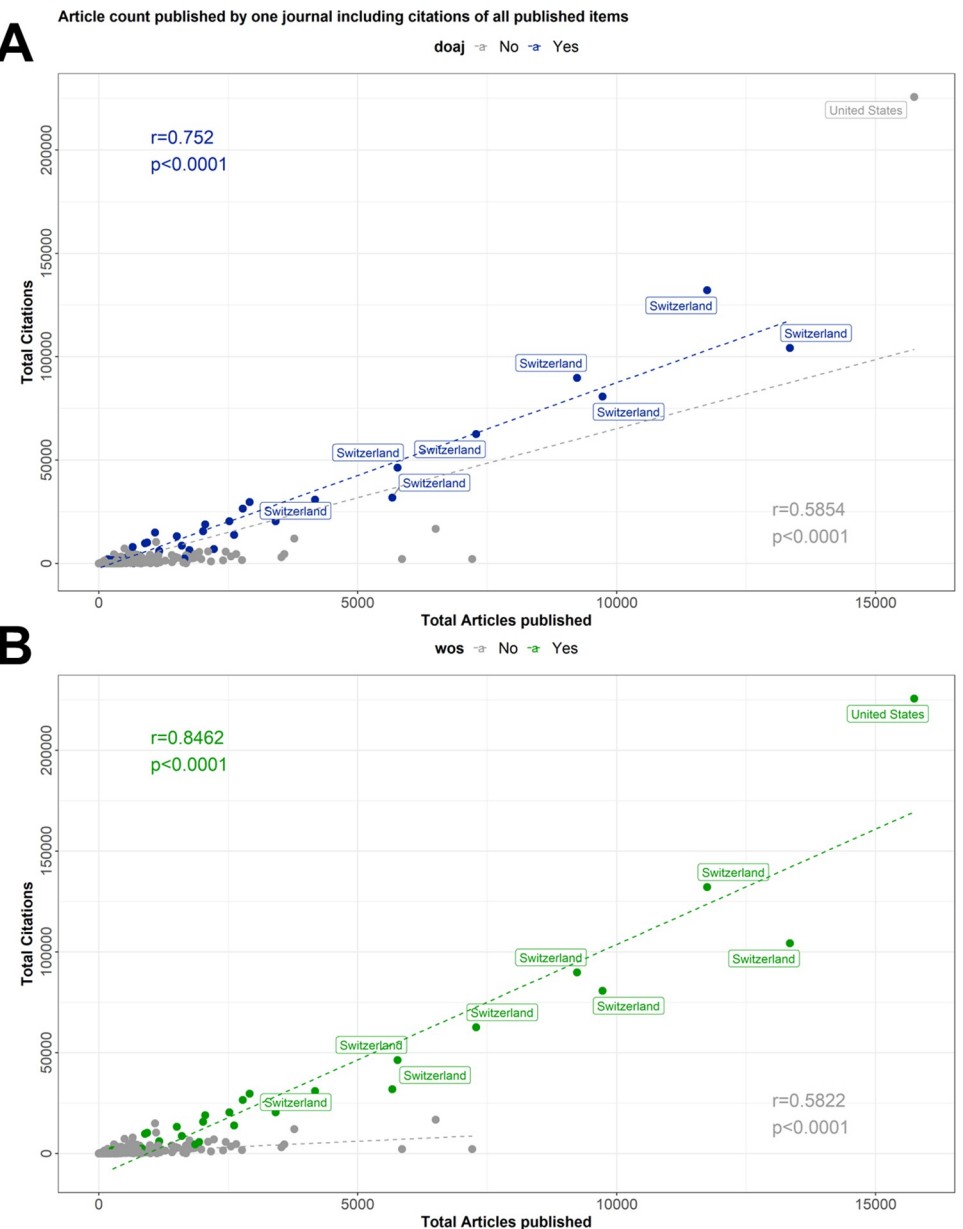

**Fig 7. Correlation between total citations and total published articles by journals listed on Beall's list in different internationally recognized databases.** Scatter plot of journals' total citations and total issued DOIs from 2011 to 2018 divided by being listed in the DOAJ (**A**) and Web of Science (**B**). Country information of the ISSN database was combined with the DOI and citation count stored in the crossref and data stored in the different databases for journals listed on Beall's list. Linear relationships between total citations and total articles published were assessed by Kendall rank correlation.

with no citations has been significantly low (median 30.3%) in WoS listed journals, followed by journals listed in PMC (median 31.9%) (Fig 8C). Splitting Beall's list journals in PMC and WoS listed, the majority was not listed in any of these databases and this was in line with low median article citation rates (S3g Fig in S2 File).

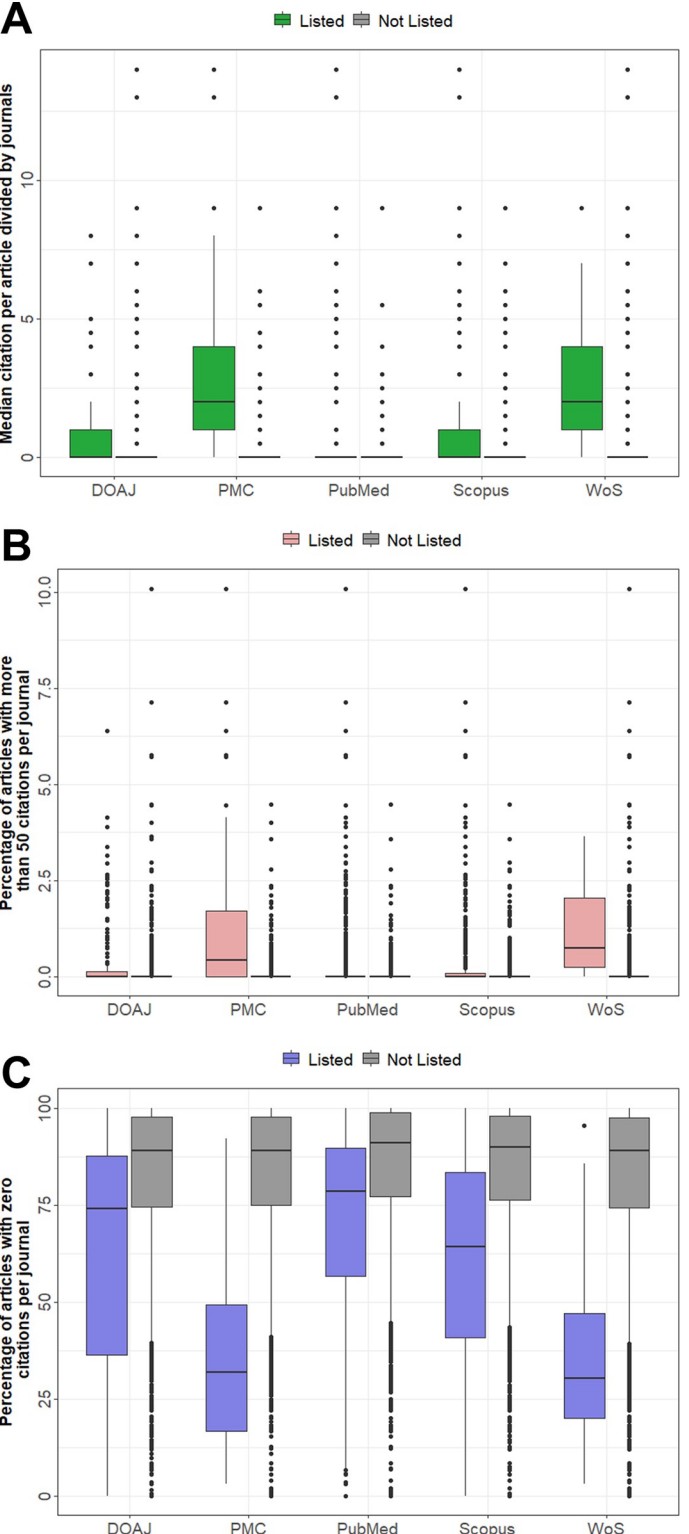

**Fig 8. Analysis of citations of articles published by journals listed on Beall's list in different internationally recognized databases.** Boxplots of journals listed in the crossref's cited-by program and **(A)** median citation per article per journal divided by database listing, **(B)** Percentage of articles with more than 50 citations per journal divided by database listing and **(C)** Percentage of articles with zero citations per journal divided by database listing. The Mann-Whitney U Test was used to compare different types of citations count and if the journal was listed in the indicated database. Data are presented as mean ± SD. * = p<0.05; ** = p<0.01; *** = p<0.001 and **** = p<0.0001.

## Discussion

Predatory journals have been discussed and investigated over the last years in many studies but due to the huge number of journals and publishers listed on Beall's list, its impact has not been properly investigated. The present study performed a large-scale bibliometric analysis of all journals listed on Beall's list and in the DOAJ to gain more insights on how the public and scientific discussion of predatory journals changed open-access publishing and on the impact on journals listed on Beall's list. However, it has to be clearly mentioned that Beall's list has been criticized by many for its methodology and has been taken offline in 2017 for this reason [15–17].

In contrast to previous studies, we demonstrate that geographic locations mentioned in the journal title often match the country information stored in the ISSN database [8]. In our study, the majority of journals were located in the United States followed by India, which is in contrast to a study published by Demir, who could demonstrate that the majority of journals on Beall's list came from India [18]. This divergence can be explained by the fact that he only used a small part of Beall's list and made a IP/WHOIS (which only gives the location of the hosting institution/VPN rather than the organization behind it), whereas we used the entry of the ISSN database to identify the country of origin.

Another important criterion of the scientific quality of a journal is its listing in databases. Several studies could demonstrate that potential predatory journals are indexed in databases like pubmed, scopus and web of science [19]. Journals can apply for inclusion by a specific database and, since different databases have different inclusion criteria, it makes a significant difference in which database a journal is listed. Since Web of Science only wants to cover the 'core journals' in a specific scientific category, each applicant is assessed by an in-house editorial team, the use of publicly available criteria, and with the help of citation analysis [20]. A similar rigorous inclusion process is used by Scopus, which is hosted by Elsevier [20]. This makes it harder for journals/publisher to be included in such databases, and, therefore, it was no surprise that the number of Beall's journals listed there was the lowest. On the other side, the highest number of journals on Beall's list was listed in crossref, which performs only a formal background check.

However, many predatory journals tend to refer to numerous databases they are listed in, although they are either not listed or they are only listed in databases with no real or poor quality controls [21]. Only a few journals on Beall's list were listed in all investigated databases, which could mean either that a high number of journals might not fulfill the scientific criteria required by the respective databases or these journals never intended to be listed (since e.g. listing costs were to high).

Prior to our study, it was unclear how many articles are published by journals on Beall's list and what has been the impact of Beall's list on open-access journals in general, but due to the systematic bibliometric big-data approach, we received by far the biggest picture of this potential problem. We demonstrated that there was a constant increase in articles published by journals on Beall's list and journals listed in the DOAJ until 2017 suggesting that authors who published in these journals may not have been aware of Beall's list or ignored their listing [22].

Regarding the first point, scientists who are scientifically active should be aware of predatory journals, but only a minority knows about Beall's list or is aware of efficient ways to reliably identify them [23,24]. In addition, there may be scientists who actively use predatory journals to publish their scientific work as fast as possible–independently of the scientific quality of the work–to fulfill the publication requirements of their universities [25]. In this study we only assessed the numbers of publications rather than the motivation of publishing in predatory journals and this can be–as mentioned above–due to multiple reasons. However, with

the multinational media coverage by several news stations in 2018, we observed a stronger decline in publishing output of journals listed on Beall's list compared to journals listed in the DOAJ [26]. One of the publishers extensively covered was the OMICS Publishing group [27]. Until 2015 there was a constant increase in articles published by the OMICS Publishing group with a plateau in 2016. Since the U.S. Federal Trade Commission has sued OMICS Publishing group [28], a decline in articles published by the OMICS Publishing group was observed from 2017 onwards. Another news initiative that probably had an impact on publishing output was the coverage of the journal *oncotarget*, *which was* suspended from being listed in Web of Science by Clarivate Analytics [29]. This has been criticized by the editor-in-chief of *oncotarget* [30], but led to a drop in submission to the journal and its publications as shown by our data. Interestingly, profiteers of 2018s situation were Frontiers Media and MDPI. Other journals on Beall's list appear to not have been negatively affected by the increase of media coverage in 2018. Frontiers media, for example, was on Beall's list at the time of analysis and increasing their article count after 2018. The same is true for MDPI, which had been on Beall's list but was later removed from it.

Another important question was if articles from journals on Beall's list actually get any citations. Therefore, we used crossref's cited-by-database to estimate the number of citations. The advantage of our study in comparison to previous studies was that there was a low entry barrier for journals/publisher to provide such information. Although it provides only a small slice of the problem, it is a larger one than in previous studies but results are still highly dependent on the information provided [31]. Importantly we could demonstrate that journals on Beall's list get indeed citations, and this was in high dependency of the databases such journals were listed in. Therefore, articles from journals listed in Web of Science were more often cited. In return —as shown by another group–even journals listed in Web of Science cite such articles [32].

## Limitations

Firstly, Beall's list has been criticized for its flawed methodology and should no longer be used to identify predatory journals. The goal of the present study was not to classify journals as predatory or non-predatory, but rather to investigate the impact of Beall's list on scientific publishing over the time period it was online. Secondly, our work is highly dependent on the quality of the databases we searched and the information gathered by the specific organization behind each database as well as information provided by the specific journal/publisher itself.

## Conclusions

Although Beall's list received broad attention by the scientific community, but it seems its impact on scientific publishing may be overestimated. This might be because Beall's list was not widely recognized, researchers rely more on other databases (e.g. Web of Science) than Beall's list or researchers ignored or did not care about the fact that journals were listed on Beall's list.

## Supporting information

**S1 File. Static lists from Beall, PMC, Web of Science, DOAJ, PubMed, Scopus and Crossref which were used in this study.**
(RAR)

**S2 File. S1 to S4 Figs supporting the data presented in this manuscript.**
(PPTX)

**S3 File. S1 to S7 Tables supporting the data presented in this manuscript.**
(DOCX)

## Acknowledgments

We gratefully thank Jenny Delasalle and Ursula Flitner for their valuable input and discussion.

## Author Contributions

**Conceptualization:** Georg Richtig, Jörg Scheffel, Marcus Maurer, Frank Siebenhaar.

**Data curation:** Georg Richtig, Marina Berger, Max Koeller, Markus Richtig, Erika Richtig.

**Formal analysis:** Georg Richtig.

**Investigation:** Georg Richtig, Marina Berger, Max Koeller, Markus Richtig, Erika Richtig, Jörg Scheffel, Marcus Maurer, Frank Siebenhaar.

**Methodology:** Georg Richtig, Jörg Scheffel, Marcus Maurer, Frank Siebenhaar.

**Project administration:** Georg Richtig, Jörg Scheffel, Marcus Maurer, Frank Siebenhaar.

**Resources:** Jörg Scheffel, Marcus Maurer, Frank Siebenhaar.

**Software:** Georg Richtig.

**Supervision:** Jörg Scheffel, Marcus Maurer, Frank Siebenhaar.

**Validation:** Georg Richtig, Jörg Scheffel, Marcus Maurer, Frank Siebenhaar.

**Visualization:** Georg Richtig.

**Writing – original draft:** Georg Richtig.

**Writing – review & editing:** Marina Berger, Max Koeller, Markus Richtig, Erika Richtig, Jörg Scheffel, Marcus Maurer, Frank Siebenhaar.

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
