## [Decision Letter · Decision Letter 0]

23 Jul 2022

PONE-D-22-08372Perception, impact and use of Beall’s list by the scientific communityPLOS ONE

Dear Dr. Richtig,

Thank you for submitting your manuscript to PLOS ONE. After careful consideration, we feel that it has merit but does not fully meet PLOS ONE’s publication criteria as it currently stands. Therefore, we invite you to submit a revised version of the manuscript that addresses the points raised during the review process.

We look forward to receiving your revised manuscript.

Kind regards,

António M. Lopes, PhD

Academic Editor

PLOS ONE

Journal Requirements:

Reviewers' comments:

Reviewer's Responses to Questions

**Comments to the Author**

1. Is the manuscript technically sound, and do the data support the conclusions?

Reviewer #1: Partly

Reviewer #2: Yes

2. Has the statistical analysis been performed appropriately and rigorously? 

Reviewer #1: No

Reviewer #2: Yes

3. Have the authors made all data underlying the findings in their manuscript fully available?

Reviewer #1: Yes

Reviewer #2: Yes

4. Is the manuscript presented in an intelligible fashion and written in standard English?

Reviewer #1: Yes

Reviewer #2: Yes

5. Review Comments to the Author

Reviewer #1: please try to incorporate the comments to give more sound for the reader/researcher when the manuscript published. The author must focus on the data explanation and he/she must use appropriate model for data analysis.

Reviewer #2: This is an interesting work. Today, many academic researchers, specifically young faculties, would like to publish their works in high impact journals for variety of reasons including promotion and reputation. The process of journal selection to be home of the new paper is rather difficult and authors sometimes do not care about the quality of the journal whether it is predatory or not and do not care about information such as Beall’s list and similar ones. They just want to publish their work somewhere. From this point of view, the investigations such as this one could be informative and useful.

Nevertheless, there are some comments that could improve the usefulness of this interesting manuscript.

1. There are a huge number of statistics in this manuscript; but the background information is not sufficiently provided. In other words, the authors need to clarify why publishing in predatory journals is problematic. If so, how the scientific community should be aware of this problem. Basically, for what reason(s) these journals were created? Is that a kind of dirty business or it is in response to some demand.

The readers of articles of this kind, would like to see some opinions on this issue. Providing statistics in not enough.

2. The authors report that some of the predatory journals on the Beall’s list are also listed in quality-driven databases such as PubMed, CrossRef, etc. The authors need to explain how these predatory journals’ names were introduced into the quality-driven databases. Is that through referencing and citation by published articles? or by other means?

3. I think it is very important to define a “predatory journal’. It is hard to believe that the publishers that publish peer-reviewed articles are also publishing non-peer-reviewed articles. For example, Frontiers Media and OMICS Publishing Group that publish high impact factor open access journals are also publishing predatory journals? If it is true, needs more clarification. How these two sorts of journals i.e. true scientific journals and predatory journals, were mixed up.

In general, I think these kinds of manuscripts have their own readership and deserve to be published after a minor revision.

6. PLOS authors have the option to publish the peer review history of their article (what does this mean?). If published, this will include your full peer review and any attached files.

Reviewer #1: **Yes: **Workineh muluken

Reviewer #2: No

---

## [Author Response · Author response to Decision Letter 0]

15 Nov 2022

Reviewer #1: please try to incorporate the comments to give more sound for the reader/researcher when the manuscript published. The author must focus on the data explanation and he/she must use appropriate model for data analysis.

Answer #1: We thank the reviewer for his/her valuable suggestions and made the following changes to the manuscript according to the reviewer’s suggestions: 

1) We further streamlined the introduction to give a more explanatory overview on why predatory journals are a serious problem for the scientific community. 

2) The discussion section has been updated to put our results in a clearer picture according to the current literature. Additionally, we rewrote the limitations section to provide readers with guidance on the interpretation of our findings and report.

3) We further included some explanatory sentences within the results section to provide focus on data explanation and context on the models of data analysis used. 

4) Concerning the use of appropriate models for data analysis, we introduced several statistical analyses and present their results. A statistical section has been added to the methods section. Furthermore, Table S2b, Table S4b, SFig 2c were added, and Fig 4a-b, Fig 5a-c and SFig 3b-f were updated. (Figure 2 has been split into Figure 2 and Figure 3. Figure 3 has been split into Figure 4 and Figure 5. Both for better readability).

5) The title has been changed from “Perception, impact and use of Beall’s list by the scientific community” to “Predatory journals: Perception, impact and use of Beall’s list by the scientific community – a bibliometric big data study”. Short title has been included.

Reviewer #2.1: 

1. There are a huge number of statistics in this manuscript; but the background information is not sufficiently provided. In other words, the authors need to clarify why publishing in predatory journals is problematic. If so, how the scientific community should be aware of this problem. Basically, for what reason(s) these journals were created? Is that a kind of dirty business or it is in response to some demand.

The readers of articles of this kind, would like to see some opinions on this issue. Providing statistics in not enough.

Answer #2.1: We thank the reviewer for this important comment. We agree with the reviewer that it is important to provide more context on why predatory journals are problematic and why they exist. Predatory journals emerged in the wake of the open access (OA) publishing model, where authors pay a fee for getting their work published and made available to the public. Quality publishing is time-consuming and costly, and some journals decided to put revenue first and quality second, set out to attract submission of as many articles as possible independent of their quality and scientific merit. 

From our perspective, there are several problems with publishing in a predatory journal. These include, but are not limited to the fact that the lack of peer-review bears the risk of publishing data that are not fully scientifically sound. Peer review, i.e. the evaluation of reports by one or more scientists with similar competencies, is an important instrument of self-regulation that helps to maintain quality standards, avoid plagiarism, and improve the quality of scientific reporting. A negative peer review should prompt journals to ask authors to improve their report or to reject the paper. This does not happen with predatory journals. Secondly, predatory journals have low or no standards with respect to the reporting of conflicts of interests, which results in the risk that such conflicts are not or not fully made transparent. Thirdly, many predatory journals exist for only a short period of time, after which their content and published reports are no longer available, resulting in loss of information. 

In response to your question and suggestion, we have now included additional wording on predatory journals and why they exist and why they are problematic:

There are several problems with publishing in a predatory journal. These include, but are not limited to the fact that the lack of peer-review bears the risk of publishing data that are not fully scientifically sound. Peer review, i.e. the evaluation of reports by one or more scientists with similar competencies, is an important instrument of self-regulation that helps to maintain quality standards, avoid plagiarism, and improve the quality of scientific reporting. A negative peer review should prompt journals to ask authors to improve their report or to reject the paper. This does not happen with predatory journals. Secondly, predatory journals have low or no standards with respect to the reporting of conflicts of interests, which results in the risk that such conflicts are not or not fully made transparent. Thirdly, many predatory journals exist for only a short period of time, after which their content and published reports are no longer available, resulting in loss of information.2

Reviewer #2.2: 2. The authors report that some of the predatory journals on the Beall’s list are also listed in quality-driven databases such as PubMed, CrossRef, etc. The authors need to explain how these predatory journals’ names were introduced into the quality-driven databases. Is that through referencing and citation by published articles? or by other means?

Answer #2.2: There are two types of databases available for science: Quality-driven database and others. In quality-driven databases the journal must meet some content criteria that go beyond formal criteria. The most rigorous criteria - in terms of form and content - were set by Web of science followed by MEDLINE. It is important to know that one does not necessarily have to be listed in MEDLINE to appear in a Pubmed search. Pubmed has also a) changed its criteria after the first large study and b) also issued a statement! (https://grants.nih.gov/grants/guide/notice-files/not-od-18-011.html) In our study, we also show quite clearly that the number of journals that were listed in Beall's list appeared relatively rarely in Web of Science in contrast to databases that do not do a content check but only formal background check (e.g. Crossref). Journals apply for inclusion in a database and have to meet only formal criteria (e.g. crossref) or also content criteria (e.g. Web of science). 

We inserted the following phrase to the discussion to clarify this issue: 

“Another important criterion of the scientific quality of a journal is its listing in databases. Journals can apply for inclusion by a specific database. Since different databases have different inclusion criteria, it makes a significant difference in which database a journal is listed. Web of Science has the most rigorous criteria for inclusion in their database, and the number of Beall’s journals listed there was the lowest. On the other side, the highest number of journals on Beall’s list was listed in crossref, which performs only a formal background check. “

The methodological differentiation between quality-driven and non-quality-driven is now stated in the methods section: 

“Databases were categorized as quality-driven or non-quality-driven databases depending on their journal inclusion criteria. If journals only had to fulfil formal criteria to be included in the database, the database was categorized as non-quality-driven (e.g. ISSN database, Crossref). If as database performs substantial background checks prior to inclusion of a journal, the database was categorized as quality-driven (e.g. WoS, PMC, Pubmed, Scopus, DOAJ).“

Reviewer #2.3: 3. I think it is very important to define a “predatory journal’. It is hard to believe that the publishers that publish peer-reviewed articles are also publishing non-peer-reviewed articles. For example, Frontiers Media and OMICS Publishing Group that publish high impact factor open access journals are also publishing predatory journals? If it is true, needs more clarification. How these two sorts of journals i.e. true scientific journals and predatory journals, were mixed up.

Answer #2.3: The definition of a “predatory journal” is rather straightforward and was reviewed by several authors in more detail including our group (Richtig et al. JEADV 2018; Beall JOSPT 2017). Briefly, a predatory journal accepts articles for publication, along with authors fees, without checks for quality, plagiarism, or ethical approval. The main challenge and the real underlying problem are how to identify “predatory journals”. In essence, whether a journal is “predatory” or not depends on its aims, intentions, and processes, which often are not made known to the general public. With non-predatory journals, peer review is the rule and publication of a paper without peer-review is a mistake that should be detected and avoided. With predatory journals, absence of peer-review or very poor review are systematic and intended (or at tolerated without intention to change this practice). Without precise knowledge of a journal’s internal structures and practices, one cannot make a statement about whether a journal is "predatory" (Discussed in Richtig et al. JEADV 2018). Also, a "predatory journal", over time, can become a non predatory one. A new open access journal with an inexperienced editor and publishing team may be viewed as "predatory" because of how it works, but has a genuine interest in becoming a “good” journal, and does so over time. Many lists of “predatory” journals readily include journals but rarely drop them (compare Figure 2 Richtig et al. JEADV 2018). Taken together, the problem is complex and was not the focus of our study, which aimed to assess how a very prominent list of predatory journals (Beall's list was the most prestigious and widely discussed) was used by the scientific community. 

To address the reviewer’s feedback, we updated the section on limitations: 

“Firstly, Beall’s list has been criticized for its flawed methodology and should no longer be used to identify predatory journals. The goal of the present study was not to classify journals as predatory or non-predatory, but rather to investigate the impact of Beall’s list on scientific publishing over the time period it was online.“

Editor comment #3.1: 

Answer #3.1: We thank the Editor for this advice. We have reviewed and adopted our manuscript to meet PLOS ONE's style requirements. 

Editor comment #3.2: 

We note that you have indicated that data from this study are available upon request. PLOS only allows data to be available upon request if there are legal or ethical restrictions on sharing data publicly. For more information on unacceptable data access restrictions, please see http://journals.plos.org/plosone/s/data-availability#loc-unacceptable-data-access-restrictions. 

Answer #3.2: Although all lists that are needed to reproduce this study can be downloaded by using wayback.org, we made a zip file with all text files to reproduce this study. This contains all lists from the methods section: “First, complete lists of journals indexed by the following databases were obtained: Scopus (downloaded from Elsevier’s Scopus database; https://www.scopus.com; 18.06.2018), Web of Science (WoS; downloaded from Clarivate Analytics; Journal Citation Reports 2017; https://jcr.clarivate.com; 12.04.2018), the Directory of Open-Access Journals (DOAJ; https://doaj.org/csv; 11.04.2018); PubMed (acquired from the U.S. National Institutes of Health (NIH; ftp://ftp.ncbi.nih.gov/pubmed/J_Medline.txt; 15.12.2018), and the U.S. National Library of Medicine’s list of PubMed Central (PMC) participating journals (https://www.ncbi.nlm.nih.gov/pmc/journals; 15.12.2018)“.

1.) bealls_journals.csv (the used version of Beall’s List from the homepage of Beall)(journals)

2.) bealls_publisher.csv (the used version of Beall’s List from the homepage of Beall)(publisher)

3.) predJournals_final4_15092018_cleaned_double.csv

4.) publisher_only_from14082017.csv

5.) PubMed: J_Medline_20181215.txt

6.) PMC: jlist_20181215.csv

7.) Web of Science: Journal_Citation_report_12042018.csv

8.) DOAJ: doaj_20180411_1930_utf8.csv

9.) Scopus: ext_list_April_2018_2017_Metrics_18062018_cleaned.csv

10.) Crossref: All Dois found published by journals on bealls list: all_dois.txt

The whole dataset contains about 250 GB of plain text and it would, therefore, be too large for complete upload. However, with the dataset provided it should be possible to reproduce the study where static databases have been used (except crossref whereas the citation data has been obtained and should be continuously updated by crossref itself).

---

## [Decision Letter · Decision Letter 1]

22 Dec 2022

PONE-D-22-08372R1Predatory journals: perception, impact and use of Beall’s list by the scientific community – a bibliometric big data studyPLOS ONE

Dear Dr. Richtig,

Thank you for submitting your manuscript to PLOS ONE. After careful consideration, we feel that it has merit but does not fully meet PLOS ONE’s publication criteria as it currently stands. Therefore, we invite you to submit a revised version of the manuscript that addresses the points raised during the review process.

We look forward to receiving your revised manuscript.

Kind regards,

António M. Lopes, PhD

Academic Editor

PLOS ONE

Reviewers' comments:

Reviewer's Responses to Questions

**Comments to the Author**

1. If the authors have adequately addressed your comments raised in a previous round of review and you feel that this manuscript is now acceptable for publication, you may indicate that here to bypass the “Comments to the Author” section, enter your conflict of interest statement in the “Confidential to Editor” section, and submit your "Accept" recommendation.

Reviewer #2: All comments have been addressed

Reviewer #3: (No Response)

2. Is the manuscript technically sound, and do the data support the conclusions?

Reviewer #2: Yes

Reviewer #3: Partly

3. Has the statistical analysis been performed appropriately and rigorously? 

Reviewer #2: Yes

Reviewer #3: I Don't Know

4. Have the authors made all data underlying the findings in their manuscript fully available?

Reviewer #2: Yes

Reviewer #3: Yes

5. Is the manuscript presented in an intelligible fashion and written in standard English?

Reviewer #2: Yes

Reviewer #3: Yes

6. Review Comments to the Author

Reviewer #2: (No Response)

Reviewer #3: It is a methodologically correct original study and its biggest advantage is the scale of the analysis that operates on a high number of journals. However, the way in which the study is described in the wider context of research on predatory publishing is sometimes unclear or misleading. Because of that, I recommend a major revision of the paper which will be focused on the Introduction and Discussion part of the paper. Below I present concrete suggestions for the revision of the paper.

1. There is a need for clarification of what kind of Beall’s list’s „impact” authors aim to study. Eg. developing policies, hiring processes, national research evaluation systems, or discourse around OA are areas where the impact of Beall’s list can be measured. The authors seem to be focused only on very general publishing patterns of researchers - which is a valid aim of the study but should be stated more clearly. This publication could be helpful for authors in expanding the context of this impact:

Krawczyk, F., & Kulczycki, E. (2021). How is open access accused of being predatory? The impact of Beall’s lists of predatory journals on academic publishing. The Journal of Academic Librarianship, 47(2), 1–11. https://doi.org/10.1016/j.acalib.2020.102271

2. Although they are right that: "Most studies performed on Beall’s list have limited their analyses either to only its listed journals, excluding listed publishers, or they investigated only a sample of the complete list.(5)", authors should better point out the originality of their research. The way they do it on Page 5 is not sufficient.

Writing about „most of the studies” is vague and they are not mentioning studies that are actually using both lists of journals and of publishers. Eg.:

Crawford, W. (2014). Journals, “Journals” and Wannabes: Investigating The List. Cites & Insights, 14(7).

Nelson, N., & Huffman, J. (2015). Predatory Journals in Library Databases: How Much Should We Worry? The Serials Librarian, 69(2), 169–192. https://doi.org/10.1080/0361526X.2015.1080782

Moreover, the authors mention that it is unknown how many articles from predatory journals "are referenced by other publications” but they do not mention any existing study of citations to predatory journals and limitations of such studies which their study would overcome. See, for instance:

Frandsen, T. F. (2017). Are predatory journals undermining the credibility of science? A bibliometric analysis of citers. Scientometrics, 113(3), 1513–1528. https://doi.org/10.1007/s11192-017-2520-x

Kulczycki, E., Hołowiecki, M., Taşkın, Z., & Krawczyk, F. (2021). Citation patterns between impact-factor and questionable journals. Scientometrics. https://doi.org/10.1007/s11192-021-04121-8

The method used by the authors also seems to have some limitations from this perspective since they are able to calculate citations to only a small part of journals from the list (n = 812).

3. In my opinion, there is also a need to revise the first paragraph of the paper (page 1). The way in which Open Access is described is incorrect. Authors write: "In the open access (OA) model, authors pay an article-processing fee for the publishing services of a journal, and everyone can read the article." For a good description of the diversity of different modes of publishing in OA, I can recommend the works of Peter Suber, for instance: Suber, P. (2012). Open Access. MIT Press. http://mitpress.mit.edu/sites/default/files/titles/content/9780262517638_Open_Access_PDF_Version.pdf

4. Moreover, describing journals financed mostly by subscriptions as „traditional” is problematic. When one looks at the history of academic publishing one can argue that a strong tradition of financing academic publishing mostly by philanthropy and scholarly societies exists. (Fyfe, A., Moxham, N., McDougall-Waters, J., & Røstvik, Camilla M. (2022). A History of Scientific Journals. UCL Press. https://doi.org/10.14324/111.9781800082328)

5. There is a need for more clarity in the section "Beall’s list did not significantly impact the Open Access movement”. The one issue is that it is not exactly clear when authors describe in this section journals from Beall’s list and when open access journals from DOAJ. The second issue is connected to a very specific understanding of impact - the authors are studying only very general publishing patterns of academics, but writing about „impacting Open access movement” may suggest that studied qualitatively how OA is perceived (eg. Scholars can publish in journals they dislike to fulfill the number of publications university require from them).

6. I believe better arguments and a reference are needed to support the statement that "Web of Science has the most rigorous criteria for inclusion in their database” on page 21. It is not entirely clear what „rigorous” mean in this context - eg. DOAJ is listing only open access journals and WoS is open also for subscription-based journals.

7. The article would have benefited if its results would be put in the wider context of the other research and their limitations in the discussion section. Eg. how authors’ interesting analysis of titles journals and locations of journals can be compared to the "Comparison of journal website contact location claims and actual locations based on the IP/WHOIS” conducted in Demir, S. B. (2018). Predatory journals: Who publishes in them and why? Journal of Informetrics, 12(4), 1296–1311. https://doi.org/10.1016/j.joi.2018.10.008

7. PLOS authors have the option to publish the peer review history of their article (what does this mean?). If published, this will include your full peer review and any attached files.

Reviewer #2: No

Reviewer #3: No

---

## [Author Response · Author response to Decision Letter 1]

6 Feb 2023

PONE-D-22-08372.R1

Dear Dr. Lopes, February 2023

we thank the reviewers for their valuable comments and the Editorial Board for the consideration of our work for possible publication in PLOS One.

The following changes have been made according to the suggestions of Reviewer #3:

Comment #1: There is a need for clarification of what kind of Beall’s list’s „impact” authors aim to study. Eg. developing policies, hiring processes, national research evaluation systems, or discourse around OA are areas where the impact of Beall’s list can be measured. The authors seem to be focused only on very general publishing patterns of researchers - which is a valid aim of the study but should be stated more clearly. This publication could be helpful for authors in expanding the context of this impact: 

Krawczyk, F., & Kulczycki, E. (2021). How is open access accused of being predatory? The impact of Beall’s lists of predatory journals on academic publishing. The Journal of Academic Librarianship, 47(2), 1–11. https://doi.org/10.1016/j.acalib.2020.102271

Answer #1: We gratefully thank the reviewer for this valuable suggestion and added the following specification to the introduction to make the aim of our study more clear in accordance to the suggested literature: “The aim of this paper is to characterize Beall’s list in total, to see where such journals/publisher came from and in which databases such journals/publisher were listed since such metrics are widely used by authors to decide where to publish. Secondly, we aimed to see how journals listed on Beall’s list performed – in terms of article count – in comparison to journals being listed in the DOAJ over the recent years. And thirdly, we investigate the number of citations that articles of journals being listed on Beall’s list received.”

Comment #2: Although they are right that: "Most studies performed on Beall’s list have limited their analyses either to only its listed journals, excluding listed publishers, or they investigated only a sample of the complete list.(5)", authors should better point out the originality of their research. The way they do it on Page 5 is not sufficient.

Writing about „most of the studies” is vague and they are not mentioning studies that are actually using both lists of journals and of publishers. Eg.:

Crawford, W. (2014). Journals, “Journals” and Wannabes: Investigating The List. Cites & Insights, 14(7).

Nelson, N., & Huffman, J. (2015). Predatory Journals in Library Databases: How Much Should We Worry? The Serials Librarian, 69(2), 169–192. https://doi.org/10.1080/0361526X.2015.1080782

Answer #2. We thank the reviewer for this supportive comment and revised the introduction section accordingly: “The majority of studies performed on Beall’s list have limited their analyses either to only its listed journals, excluding listed publishers, investigated only a sample of the complete list or used one of the earlier versions of the list.(8,12) Although there has been some work done in analysing citations of articles from potential predatory journals, these analyses were limited to databases which had quality driven inclusion criteria.(12)”

Comment #3: Moreover, the authors mention that it is unknown how many articles from predatory journals "are referenced by other publications” but they do not mention any existing study of citations to predatory journals and limitations of such studies which their study would overcome. See, for instance:

Frandsen, T. F. (2017). Are predatory journals undermining the credibility of science? A bibliometric analysis of citers. Scientometrics, 113(3), 1513–1528. https://doi.org/10.1007/s11192-017-2520-x

Kulczycki, E., Hołowiecki, M., Taşkın, Z., & Krawczyk, F. (2021). Citation patterns between impact-factor and questionable journals. Scientometrics. https://doi.org/10.1007/s11192-021-04121-8

The method used by the authors also seems to have some limitations from this perspective since they are able to calculate citations to only a small part of journals from the list (n = 812).

Answer #3: We agree with the reviewer that we haven’t clearly pointed out the potential advantages of our study and how it adds to previous analyses. To address this concern, we added wording on this to the limitation section of the discussion and highlighted the challenges of big data bibliometric studies: “Prior to our study, it was unclear how many articles are published by journals on Beall’s list and what has been the impact of Beall’s list on open-access journals in general, but due to the systematic bibliometric big-data approach, we received by far the biggest picture of this potential problem. We demonstrated that there was a constant increase in articles published by journals on Beall’s list and journals listed in the DOAJ until 2017 suggesting that authors who published in these journals may not have been aware of Beall’s list or ignored their listing.” Furthermore, we included the above mentioned studies into the discussion.

Comment #4: In my opinion, there is also a need to revise the first paragraph of the paper (page 1). The way in which Open Access is described is incorrect. Authors write: "In the open access (OA) model, authors pay an article-processing fee for the publishing services of a journal, and everyone can read the article." For a good description of the diversity of different modes of publishing in OA, I can recommend the works of Peter Suber, for instance: Suber, P. (2012). Open Access. MIT Press. http://mitpress.mit.edu/sites/default/files/titles/content/9780262517638_Open_Access_PDF_Version.pdf

Answer #4: We thank the reviewer for this well-taken point and agree that there are nowadays several different Open-access models (green OA, gold OA, hybrid OA,…) available but we tried to break it down and simplify the model for an easier understanding of the problem although – from a publishing point of view – it is not fully correct. We aimed to focus on the basic idea behind the open-access model in more detail in the introduction according to the review’s suggestion.

Comment #5. Moreover, describing journals financed mostly by subscriptions as „traditional” is problematic. When one looks at the history of academic publishing one can argue that a strong tradition of financing academic publishing mostly by philanthropy and scholarly societies exists. (Fyfe, A., Moxham, N., McDougall-Waters, J., & Røstvik, Camilla M. (2022). A History of Scientific Journals. UCL Press. https://doi.org/10.14324/111.9781800082328)

Answer #5. We thank the reviewer for raising this point, it is well taken. We completely reworded the first paragraph, bringing on board the suggestions of the reviewer and tried to summarize this complicated topic in a short overview. 

“In the last decades, authors submitted their work for peer-review to be published in a paper-based journal. Costs were covered by a scientific society behind the journal, subscription fees to the journal, and/or industry sponsoring.(1) Importantly, access to articles is restricted to subscribers and those who pay per article. In the open-access (OA) model, articles are freely available to everyone who wants to read them.(2) However, the most common model of open-access is that authors pay a publisher an article-processing fee for the publishing services of a journal – including expenses for editing, distributing, hosting the article and peer-reviewing an article – and therefore for the option to make their article globally available at no cost for the reader. The transition of the open-access model in science has been performed in many different ways with many different OA models including green OA, Diamond OA, Gold OA, Bronze OA and many more.(3)“

Comment #6. There is a need for more clarity in the section "Beall’s list did not significantly impact the Open Access movement”. The one issue is that it is not exactly clear when authors describe in this section journals from Beall’s list and when open access journals from DOAJ. The second issue is connected to a very specific understanding of impact - the authors are studying only very general publishing patterns of academics, but writing about „impacting Open access movement” may suggest that studied qualitatively how OA is perceived (eg. Scholars can publish in journals they dislike to fulfill the number of publications university require from them).

Answer #6. We thank the reviewer for raising this very important point. The aim of this section was to see if the discussion around Beall's list and the open-access movement had a general impact on the open-access model. A negative association with the open-access movement should also have led to a reduced article count and therefore harmed the idea of the open-access movement. We tried to clarify this section by providing a more clear introduction section within the paragraph: “To assess the impact of Beall’s list on Open-Access publishing as well as on journals’ publishing output of journals on Beall’s list, we compared the article count of journals on Beall’s list in comparison to journals on the DOAJ.”

We further added the following sentence to the discussion section to further highlight the problem of predatory journals and our study’s limitations. 

“In addition, there may be scientists who actively use predatory journals to publish their scientific work as fast as possible – independently of the scientific quality of the work – to fulfill the publication requirements of their universities.(24) In this study we only assessed the numbers of publications rather than the motivation of publishing in predatory journals and this can be – as mentioned above – due to multiple reasons.“

Comment #7. I believe better arguments and a reference are needed to support the statement that "Web of Science has the most rigorous criteria for inclusion in their database” on page 21. It is not entirely clear what „rigorous” mean in this context - eg. DOAJ is listing only open access journals and WoS is open also for subscription-based journals.

Answer #7. Thank you for this helpful feedback. We refined our statement in the manuscript and included more detailed information on inclusion criteria in Web of Science and why they were more “rigorous”. The paragraph was reworded and expanded to: 

“Another important criterion of the scientific quality of a journal is its listing in databases. Several studies could demonstrate that potential predatory journals are indexed in databases like pubmed, scopus and web of science.(18) Journals can apply for inclusion by a specific database and, since different databases have different inclusion criteria, it makes a significant difference in which database a journal is listed. Since Web of Science only wants to cover the ‘core journals’ in a specific scientific category, each applicant is assessed by an in-house editorial team, the use of publicly available criteria, and with the help of citation analysis.(19) A similar rigorous inclusion process is used by Scopus, which is hosted by Elsevier.(19) This makes it harder for journals/publisher to be included in such databases, and, therefore, it was no surprise that the number of Beall’s journals listed there was the lowest.“

Comment #8. The article would have benefited if its results would be put in the wider context of the other research and their limitations in the discussion section. Eg. how authors’ interesting analysis of titles journals and locations of journals can be compared to the "Comparison of journal website contact location claims and actual locations based on the IP/WHOIS” conducted in Demir, S. B. (2018). Predatory journals: Who publishes in them and why? Journal of Informetrics, 12(4), 1296–1311. https://doi.org/10.1016/j.joi.2018.10.008

Answer #8. We are grateful for these very valuable comments and have incorporated the suggested studies in the discussion section and also elaborated on limitations and differences in the context of our study. We have revised several paragraphs in our manuscript to provide more detailed information on the broader context of our study compared to other research. We truly believe that the valuable comments and suggestions given by the reviewer helped to critically improve our manuscript and to facilitate the digestion of the topic by the reader. 

“In our study, the majority of journals were located in the United States followed by India, which is in contrast to a study published by Demir, who could demonstrate that the majority of journals on Beall’s list came from India.(17) This divergence can be explained by the fact that he only used a small part of Beall’s list and made a IP/WHOIS (which only gives the location of the hosting institution/VPN rather than the organization behind it), whereas we used the entry of the ISSN database to identify the country of origin.”

“Since Web of Science only wants to cover the ‘core journals’ in a specific scientific category, each applicant is assessed by an in-house editorial team, the use of publicly available criteria, and with the help of citation analysis.(19) A similar rigorous inclusions process is used by Scopus, which is hosted by Elsevier.(19) This makes it harder for journals/publisher to be included in such databases, and, therefore, it was no surprise that the number of Beall’s journals listed there was the lowest”

“Another important question was if articles from journals on Beall’s list actually get any citations. Therefore, we used crossref’s cited-by-database to estimate the number of citations. The advantage of our study in comparison to previous studies was that there was a low entry barrier for journals/publisher to provide such information. Although it provides only a small slice of the problem, it is a larger one than in previous studies, but results are still highly dependent on the information provided.(29) Importantly we could demonstrate that journals on Beall’s list get indeed citations, and this was in high dependency of the databases such journals were listed in. Therefore, articles from journals listed in Web of Science were more often cited. In return - as shown by another group – even journals listed in Web of Science cite such articles.(30)”

---

## [Decision Letter · Decision Letter 2]

26 Apr 2023

PONE-D-22-08372R2Predatory journals: perception, impact and use of Beall’s list by the scientific community – a bibliometric big data studyPLOS ONE

Dear Dr. Richtig,

Thank you for submitting your manuscript to PLOS ONE. After careful consideration, we feel that it has merit but does not fully meet PLOS ONE’s publication criteria as it currently stands. Therefore, we invite you to submit a revised version of the manuscript that addresses the points raised during the review process.

We look forward to receiving your revised manuscript.

Kind regards,

António M. Lopes, PhD

Academic Editor

PLOS ONE

Additional Editor Comments:

Due to its controversial potential, the manuscript needs to be revised by clarifying some aspects. Please see bellow.

Main comment:

"The manuscript's conclusions are overstated in some cases. As an observational study the authors need to apply caution in inferring causation. We don't know what would have happened to the titles in Beall's list had they not been included, so we can't really include anything about the direct impact of inclusion in the list. Revisions are required to address, as follows:

Line 47: "Beall’s list, for the scientific community, appears not to be crucial for publishing in journals or for citing their content."

Line 345: "Beall’s list did not significantly impact the Open-Access movement." - needs to be toned down - needs to report result here (that articles list in Beall's list show similar publishing patterns to those not included in the list) Line 548: "Although there was and is a substantial discussion around Beall’s list and predatory journals, this seems to have had little impact on scientific publishing itself"

Other notes:

The section 'Beall’s list did not significantly impact the Open-Access movement' talks about journals being successful and unsuccessful. This language is ambiguous (a journal could be successful by reducing its published output, but increasing standards) and needs to be refined.

Lines 526-529 are unclear, specifically "the first was and the latter has been on Beall’s list" - aren't these the same thing?

The text in figures is very small - font size should be increased "

Reviewers' comments:

Reviewer's Responses to Questions

**Comments to the Author**

1. If the authors have adequately addressed your comments raised in a previous round of review and you feel that this manuscript is now acceptable for publication, you may indicate that here to bypass the “Comments to the Author” section, enter your conflict of interest statement in the “Confidential to Editor” section, and submit your "Accept" recommendation.

Reviewer #2: All comments have been addressed

Reviewer #3: All comments have been addressed

2. Is the manuscript technically sound, and do the data support the conclusions?

Reviewer #2: Yes

Reviewer #3: Yes

3. Has the statistical analysis been performed appropriately and rigorously? 

Reviewer #2: Yes

Reviewer #3: I Don't Know

4. Have the authors made all data underlying the findings in their manuscript fully available?

Reviewer #2: Yes

Reviewer #3: Yes

5. Is the manuscript presented in an intelligible fashion and written in standard English?

Reviewer #2: Yes

Reviewer #3: Yes

6. Review Comments to the Author

Reviewer #2: My comments have already been answered in the previous round of revision. I think this manuscript answers some the key questions that many young scientists and new authors are routinely facing. I recommend the publication of the manuscript and hopefully it will serve the research community.

Reviewer #3: I think that all my comments were answered very well, and article is now clearly presents new analysis of Beall's lists. I am sure that this study adds new insight in the context of the previous studies on the topic and can be published in Plos One.

When tracing the changes I spotted only one fragment that needs minor correction. The citation in the last sentence of the paragraph on page 5 seems to be incorrect - Crawford uses both lists of journals but do not provide citation analysis.

"Although there has been some work done in analysing citations of articles from potential predatory journals, these analyses were limited to databases which had quality driven inclusion criteria.(12)"

Moreover, after revision the authors could also revise the beginning of the next paragraph for stylistic purposes.

"It remains, therefore, unknown how many journals on Beall’s list are included in

quality-driven databases such as PubMed, how many articles are published by the

journals on Beall’s list and how this number has changed over the past years, or how

often articles published by journals on Beall’s list are referenced by other publications."

In the current version those two mentions of "quality driven" in almost opposite context one after another is confusing.

7. PLOS authors have the option to publish the peer review history of their article (what does this mean?). If published, this will include your full peer review and any attached files.

Reviewer #2: No

Reviewer #3: No

---

## [Author Response · Author response to Decision Letter 2]

25 May 2023

PONE-D-22-08372R3

Dear Dr. Lopes, 23th May 2023

we thank the reviewers for their valuable comments and the Editorial Board for the consideration of our work for possible publication in PLOS One.

The following changes have been made according to the suggestions of all reviewers:

Academic Editor:

Due to its controversial potential, the manuscript needs to be revised by clarifying some aspects. Please see bellow.

"The manuscript's conclusions are overstated in some cases. As an observational study the authors need to apply caution in inferring causation. We don't know what would have happened to the titles in Beall's list had they not been included, so we can't really include anything about the direct impact of inclusion in the list.

Answer #1: Thanks for raising this point, it is well taken. We revised the manuscript accordingly, bringing your suggestions on board.

Revisions are required to address, as follows:

Comment #1.1: Line 47: "Beall’s list, for the scientific community, appears not to be crucial for publishing in journals or for citing their content."

Answer #1.1: Thanks for the suggestion. We reworded the sentence: „It seems that the importance of Beall’s list for the scientific community is overestimated.“

Comment #1.2: Line 345: "Beall’s list did not significantly impact the Open-Access movement." - needs to be toned down - needs to report result here (that articles list in Beall's list show similar publishing patterns to those not included in the list) 

Answer #1.2: We thank the reviewer for his comment and changed the sentence from „Beall’s list did not significantly impact the Open-Access movement“ to „ Articles published in journals in Beall's list show similar publishing patterns to those in journals not included in the list, suggesting that Beall’s list had no major influence on the Open-Access movement“. 

The comparison of Beall’s list with the DOAJ starts with Figure 2, where there is a direct comparison of both publication numbers, followed by subsequent analysis of both lists.

Comment #1.3: Line 548: "Although there was and is a substantial discussion around Beall’s list and predatory journals, this seems to have had little impact on scientific publishing itself"

Answer #1.3: Line 548 „Although there was and is a substantial discussion around Beall’s list and predatory journals, this seems to have had little impact on scientific publishing itself“ has been changed to „Although Beall’s list received broad attention by the scientific community, but it seems its impact on scientific publishing may be overestimated.“

Comment #1.4: The section 'Beall’s list did not significantly impact the Open-Access movement' talks about journals being successful and unsuccessful. This language is ambiguous (a journal could be successful by reducing its published output, but increasing standards) and needs to be refined.

Answer #1.4: We agree with this comment and reworded this section accordingly. Specifically, the section now reads: “When the journals’ annual output was more closely analysed over the past seven years, 52 journals (0.7%) increased the number of published articles from year to year (median: 3), and no journal had a continuous decrease over the entire seven year period. When journals were grouped by their publishers, 104 of 2,296 (4.5%) publishers (287 journals) had a continuous increase in published articles over the six (or seven) year period. Within the DOAJ listed journals, MDPI was the publisher with 74 journals with a continuous increase in published articles over this time, followed by Springer (Biomed Central Ltd.) with 21 journals (Fig. 3c and Table S5).”

Comment #1.5: Lines 526-529 are unclear, specifically "the first was and the latter has been on Beall’s list" - aren't these the same thing?

Answer #1.5: Thank you for the question. Frontiers was and still is on Beall’s list. In contrast, MDPI was initially included in Beall's list, but it was later removed as MDPI effectively contested Beall's inclusion. To clarify this we reworded the sentence: „Other journals on Beall’s list appear to not have been negatively affected by the increase of media coverage in 2018. Frontiers media, for example, was on Beall’s list at the time of analysis and increasing their article count after 2018. The same is true for MDPI, which had been on Beall’s list but was later removed from it.“

Comment #1.6: The text in figures is very small - font size should be increased "

Answer #1.6: We thank the reviewer for this helpful suggestion and revised the figures 1 to 4, which are now labeled as Figures 1 to 8 to improve readability additionally to the increased font size in each figure.

Reviewer #3:

Comment #2.1: When tracing the changes I spotted only one fragment that needs minor correction. The citation in the last sentence of the paragraph on page 5 seems to be incorrect - Crawford uses both lists of journals but do not provide citation analysis.

"Although there has been some work done in analysing citations of articles from potential predatory journals, these analyses were limited to databases which had quality driven inclusion criteria.(12)"

Answer #2.1: We thank the reviewer for this important observation and included now the correct study: Björk, B.-C.; Kanto-Karvonen, S.; Harviainen, J.T. How Frequently Are Articles in Predatory Open Access Journals Cited. Publications 2020, 8, 17. https://doi.org/10.3390/publications8020017

Comment #2.2: Moreover, after revision the authors could also revise the beginning of the next paragraph for stylistic purposes.

"It remains, therefore, unknown how many journals on Beall’s list are included in

quality-driven databases such as PubMed, how many articles are published by the

journals on Beall’s list and how this number has changed over the past years, or how

often articles published by journals on Beall’s list are referenced by other publications."

In the current version those two mentions of "quality driven" in almost opposite context one after another is confusing.

Answer #2.2: We agree with the reviewer and changed the phrase before the paragraph. „quality driven“ was replaced by „more rigorous“.

---

## [Editor Report · Decision Letter 3]

8 Jun 2023

Predatory journals: perception, impact and use of Beall’s list by the scientific community – a bibliometric big data study

PONE-D-22-08372R3

Dear Dr. Richtig,

We’re pleased to inform you that your manuscript has been judged scientifically suitable for publication and will be formally accepted for publication once it meets all outstanding technical requirements.

Kind regards,

António M. Lopes, PhD

Academic Editor

PLOS ONE
---

## [Editor Report · Acceptance letter]

16 Jun 2023

PONE-D-22-08372R3 

Predatory journals: perception, impact and use of Beall’s list by the scientific community – a bibliometric big data study 

Dear Dr. Richtig:

I'm pleased to inform you that your manuscript has been deemed suitable for publication in PLOS ONE. Congratulations! Your manuscript is now with our production department. 

Kind regards, 

on behalf of

Dr. António M. Lopes 

Academic Editor

PLOS ONE